# Innovative Strategies for Liver Transplantation: The Role of Mesenchymal Stem Cells and Their Cell-Free Derivatives

**DOI:** 10.3390/cells13191604

**Published:** 2024-09-25

**Authors:** Miho Akabane, Yuki Imaoka, Jun Kawashima, Yutaka Endo, Austin Schenk, Kazunari Sasaki, Timothy M. Pawlik

**Affiliations:** 1Department of Surgery, The Ohio State University Wexner Medical Center and James Comprehensive Cancer Center, Columbus, OH 43210, USA; miho.akabane@osumc.edu (M.A.); jun.kawashima@osumc.edu (J.K.); austin.schenk@osumc.edu (A.S.); 2Division of Abdominal Transplant, Department of Surgery, Stanford University, Stanford, CA 94305, USA; imaokay@stanford.edu (Y.I.); sasakik@stanford.edu (K.S.); 3Department of Transplant Surgery, University of Rochester Medical Center, Rochester, NY 14642, USA; yutaka_endo@urmc.rochester.edu

**Keywords:** liver transplantation, mesenchymal stem cells, extracellular vesicles

## Abstract

Despite being the standard treatment for end-stage liver disease, liver transplantation has limitations like donor scarcity, high surgical costs, and immune rejection risks. Mesenchymal stem cells (MSCs) and their derivatives offer potential for liver regeneration and transplantation. MSCs, known for their multipotency, low immunogenicity, and ease of obtainability, can differentiate into hepatocyte-like cells and secrete bioactive factors that promote liver repair and reduce immune rejection. However, the clinical application of MSCs is limited by risks such as aberrant differentiation and low engraftment rates. As a safer alternative, MSC-derived secretomes and extracellular vesicles (EVs) offer promising therapeutic benefits, including enhanced graft survival, immunomodulation, and reduced ischemia–reperfusion injury. Current research highlights the efficacy of MSC-derived therapies in improving liver transplant outcomes, but further studies are necessary to standardize clinical applications. This review highlights the potential of MSCs and EVs to address key challenges in liver transplantation, paving the way for innovative therapeutic strategies.

## 1. Introduction

Liver injury can result from various hepatotoxic factors, including drugs, viruses, autoimmune responses, and lipid deposition [1,2]. Globally, 3.5% of deaths are attributed to liver cirrhosis, liver failure, or hepatocellular carcinoma (HCC) [3]. In the setting of end-stage liver disease in which the remaining liver tissue cannot regenerate quickly enough, liver transplantation is the standard treatment. However, the availability of donors is limited, the surgical costs are high, and there is a significant risk of immune rejection [4]. These challenges necessitate the exploration of alternative therapies.

Stem cell therapy has demonstrated promise not only in stimulating tissue regeneration and liver repair but also in enhancing the success of liver transplantation. Stem cells, known for their multipotency and ability to self-renew and differentiate into various somatic cells, have garnered interest for their potential to restore tissue function in vivo with minimal immune rejection and improve transplant viability. Among these, adult stem cells, including organ-specific stem cells such as hepatic progenitor cells, offer significant potential for liver regeneration [5]. These cells can differentiate into specialized liver cells and have been shown to support liver repair. However, their regenerative capacity is more limited than embryonic stem cells (ESCs), induced pluripotent stem cells (iPSCs), and mesenchymal stem cells (MSCs) [6]. ESCs, derived from early-stage embryos, possess high differentiation potential across multiple lineages. Takahashi and Yamanaka achieved a breakthrough by reprogramming mouse fibroblasts into iPSCs using the retroviral transduction of four transcription factors [7]. These reprogrammed cells exhibit similar multipotency and self-renewal capabilities to ESCs [8]. MSCs, first discovered in bone marrow in 1976 and subsequently identified in nearly all human tissues [9,10,11,12,13,14], are favored for their higher potency and availability, lack of ethical issues compared with ESCs, and lower tumorigenicity than iPSCs [15]. Consequently, MSCs are extensively employed in tissue repair and regenerative medicine and may improve transplant outcomes [16]. Transplanted MSCs can differentiate into hepatocyte-like cells, exhibit anti-fibrotic and immunomodulatory effects, and enhance graft survival by reducing immune rejection. However, the clinical application of MSCs has been hindered by risks of aberrant differentiation, low in vivo engraftment, and potential tumorigenicity [17,18,19,20,21]. Additionally, the need for large quantities of MSCs for transplantation and the prolonged expansion time pose challenges [22]. MSC transplantation can be a high-risk clinical application as living cells may induce occlusion in the microvasculature [23,24,25]. To overcome these limitations, cell-free therapies using MSC-derived secretomes have emerged as a safer, more effective alternative. The MSC-derived secretome contains numerous bioactive molecules, including soluble proteins, lipids, nucleic acids, and extracellular vesicles (EVs), which can repair tissue injury and improve transplant outcomes [26]. These soluble factors can be isolated and manipulated akin to pharmaceutical agents, allowing for long-term storage and clinical application without the associated risks of MSC transplantation [24,27,28,29]. This review examines the mechanisms by which MSC-based cell-free strategies alleviate liver injury and how these innovations could improve outcomes in liver transplantation, addressing key challenges such as immune rejection and donor organ availability. But what are MSCs?

MSCs are multipotent stem cells capable of differentiating into three primary cell lineages: adipogenic, chondrogenic, and osteogenic [9,10,11,12,13,14,15,30]. MSCs are highly accessible for therapeutic use. MSCs offer several advantages in liver transplantation and regeneration (Figure 1) [26,31,32]. MSCs promote liver regeneration by differentiating into hepatocyte-like cells, which perform essential liver functions, such as albumin synthesis, detoxification, and glucose metabolism [33,34]. These hepatocyte-like cells are key to restoring liver function in damaged tissues. Furthermore, MSCs secrete bioactive factors that modulate immune responses, inhibit fibrosis, stimulate vascularization, and promote tissue remodeling [33]. In the context of ischemia–reperfusion injury (IRI), MSCs can reduce inflammation and cell damage, improving graft survival. For immunosuppression therapy, MSCs help prevent acute cellular rejection (ACR) and graft-versus-host disease (GVHD) by modulating T-cell responses and increasing regulatory T cells (Tregs) [32,35,36,37].

Additionally, MSC-derived secretomes and EVs offer potential in organ preservation by enhancing the vitality and function of donor organs and mitigating IRI [38]. MSC-derived secretomes and EVs also aid in the restoration of organ function by promoting tissue repair and reducing apoptosis and inflammation. These advantages position MSCs and their derivatives as promising therapeutic agents in liver transplantation, with the potential to improve outcomes and address challenges, such as organ shortages and immune rejection.

## 2. MSCs in Liver Regeneration

MSCs are emerging as a promising option for liver regeneration due to their low immunogenicity, acquisition feasibility, self-renewal capabilities, and absence of ethical concerns [31]. MSCs can differentiate into hepatocyte-like cells, aiding liver regeneration and responding to cellular damage signals by migrating to injury sites [33]. Treatment with MSCs has been demonstrated to accelerate liver regeneration after a two-thirds partial hepatectomy in mice [39]. While holding potential as an alternative to liver transplantation, cell therapies come with limitations and risks. One study demonstrated that hepatocytes derived from MSCs constitute only about 1% of the total liver mass post-transplantation [40]. To address these limitations, MSC-sourced secretomes have emerged as a promising alternative.

An MSC-conditioned medium (CM) is rich in bioactive factors such as chemokines, cytokines, immunomodulatory molecules, and growth factors that exert immunomodulatory functions, inhibit fibrosis and cell death, stimulate vascularization, promote tissue remodeling, and recruit other cells [41]. MSCs also secrete EVs, including exosomes, microvesicles, and apoptotic bodies [26]. Exosomes, nanosized EVs (30–100 nm), originate from the inward budding of the membrane of multivesicular bodies (MVBs). These vesicles are released following the fusion of MVBs with the plasma membrane and are then taken up by target cells [42,43]. Exosomes encapsulate various paracrine factors, such as miRNAs and proteins, which play roles in immune regulation, energy metabolism, antioxidative stress, and tissue regeneration [44]. The nanometer-sized features of exosomes facilitate their transfer through blood and other biological fluids, protecting active ingredients from degradation and enabling intracellular uptake via endocytosis [45,46,47].

Research has demonstrated that the catalytic capacity of exosome-derived enzymes changes according to the surrounding microenvironment, making exosomes safer than other agents [48]. Various miRNAs, such as miR148a, miR378, miR532-5p, and let-7f, are enriched in the culture medium of MSCs, which can be isolated to regulate different pathways, including cellular transport, proteolysis, angiogenesis, and apoptosis [49,50,51,52,53]. Unique proteins, such as heat shock protein 70 (HSP70) and tetraspanin CD63, have been identified in MSC-derived exosomes, distinguishing them from their parental MSCs [54,55]. Soluble factors isolated from MSCs derived from different sources also exhibit varying transcriptome and proteome profiles [56]. For instance, adipose tissue-derived MSCs demonstrate more potent immunomodulatory effects compared to bone marrow-derived MSCs [57]. While bone marrow-derived MSCs outperform adipose tissue-derived MSCs in preserving tissue viability and promoting neovascularization [58]. MSC-derived soluble factors are preferentially taken up by injured tissues and mediate cell communication in both adjacent and remote areas through endocrine and paracrine signaling.

MSC-derived secretomes and exosomes have demonstrated significant potential in treating liver injuries by promoting hepatocyte proliferation, inhibiting apoptosis, necroinflammation, oxidative stress, immune rejection, and hepatic stellate cell (HSC) activation [59]. For instance, the infusion of MSC-CM immediately before liver irradiation has been proven to reduce the apoptosis of sinusoidal endothelial cells (SECs) and mitigate histopathological changes in the irradiated liver by decreasing inflammation. In rat models of reduced-size liver transplantation, MSC-CM reduced liver injury and enhanced survival rates by lowering apoptosis rates of hepatocytes and SECs, decreasing pro-inflammatory cytokine secretion, reducing neutrophil infiltration, and suppressing Kupffer cell activation [60]. Additionally, MSC-CM increased the expression of the vascular endothelial growth factor and matrix metallopeptidase 9, promoting liver regeneration in the grafts of recipient rats [60]. MSC-CM also has potential in ameliorating acute liver failure by reducing fibrogenesis, apoptosis, necroinflammation, and HSC activation, while improving glycogen synthesis and storage. Furthermore, MSC-CM has demonstrated the ability to convert CD4+ T lymphocytes into anti-inflammatory Tregs and Th2 cells, inducing HSC death within liver tissue [61].

Although there are various causes of liver injury, MSCs exhibit distinct mechanisms of action depending on their specific etiology [62]. MSCs hold potential as a cellular therapy for patients with alcoholic liver disease by promoting differentiation, modulating the immune response, inhibiting liver fibrosis, and aiding in tissue regeneration. MSCs can differentiate into hepatic cells, helping to restore hepatocyte function lost due to alcohol consumption. They also induce the expression of hepatocyte markers, including cytokeratin 18 (CK18), CK19, and cytochrome P450 3A4, while enhancing glycogen storage and albumin secretion. Additionally, MSC therapy activates dendritic cells, lymphocytes, and Tregs [63]. MSCs also release trophic factors such as VEGF, EGF, and IGF-1, which are involved in tissue repair processes [64]. Regarding non-alcoholic steatohepatitis (NASH), recent studies have applied MSC-based therapy in mouse models that mimic the characteristics of NASH [65,66]. Specifically, male immunodeficient mice were fed a high-fat diet for 21 weeks, followed by the intrasplenic administration of 0.9–1 × 10^6^ human bone marrow-derived MSCs. This treatment reduced the liver fat deposition threefold compared to control animals. Human mitochondria from the donor cells were detected in the livers of mice post-transplantation, contributing to lipid metabolism and resulting in a 25% reduction in triglyceride levels. Tissue inflammation and fibrosis were also alleviated, leading to restored tissue homeostasis [65,66].

MSC-derived exosomes have superior efficacy compared with MSCs to alleviate liver fibrosis [67]. Exosomes from human umbilical cord-derived MSCs reduced serum levels of aspartate aminotransferase (AST) and alanine aminotransferase (ALT) and repaired injured liver tissue in an acute liver failure mouse model by reducing TXNIP/NLRP3 inflammasome activation and its downstream inflammatory factors, including caspase-1, IL-1β, and IL-6 [68]. However, challenges remain, including the need for umbilical cord donation, manufacturing standardization, product characterization, and managing inter-donor variability [69]. Exosomes have also demonstrated the dose-dependent inhibition of apoptosis in D-galactoside (D-GalN)/lipopolysaccharide (LPS)-treated AML12 cells in vitro, with transplantation inhibiting hepatocyte apoptosis and increasing survival rates in D-GalN/LPS-induced acute liver failure mice [70,71].

The therapeutic potential of MSCs can be enhanced by modifying their cell culture conditions, which affects the composition of their secretome [72,73,74]. Preconditioning strategies prepare the cells for in vivo transplantation, improving their survival rate and paracrine effects [75]. Some of these strategies involve exposing the cells to physical or environmental stressors and using pharmacological modulators. For instance, thermal preconditioning at 42 °C for 1–2 h before transplantation has been shown to promote cell survival in vivo [76]. Hypoxic preconditioning, which replicates natural oxygen levels (1–12% in vivo) as opposed to the typical 21% oxygen level in vitro, boosts MSC self-renewal abilities and multipotency [77,78].

While MSCs have shown the ability to migrate and integrate into tumor tissues [79], their effects on HCC cells remain controversial. Zhao et al. [80] found that adipose tissue-derived MSC-CM inhibited proliferation and promoted cell death in an HCC cell line in vitro. However, some research suggests that bone marrow-derived MSCs can enhance the migration and invasion of HCC cells [81,82]. Studies have also raised concerns about the genetic instability and tumorigenic potential of MSC cultures [83]. Rosland et al. [84] reported that 45.8% of human MSCs underwent malignant transformation after extended culture periods. Ren et al. [85] similarly observed that MSCs from adult cynomolgus monkeys transformed into highly tumorigenic mesenchymal cells after in vitro culture. Although long-term follow-up studies have indicated that MSC transplantation is generally safe, with no tumor formation reported in patients over 11 years and 5 months [86], it is still not clear how MSCs influence tumorigenesis and development in patients. Further research is needed to clarify MSCs’ potential role in tumor development. This highlights the need for continued evaluation and the optimization of MSC-based clinical applications.

## 3. Ischemia–Reperfusion Injury and Immunosuppression Therapy

ACR and GVHD pose challenges in liver transplantation. Recently, the application of MSCs has increased in the treatment of inflammatory disorders, including organ rejection and GVHD [32]. MSCs can suppress T-cell proliferation in response to mitogenic and alloantigenic stimulations while increasing the Treg population, which is beneficial for treating ACR [35,36,37]. Specifically for GVHD, MSCs have demonstrated promise in driving T-cell-mediated immunosuppressive activity in GVHD-specific animal models [87] and have been utilized in clinical settings to treat GVHD [88,89,90,91,92].

MSCs can mitigate the inflammatory process during IRI, which occurs when previously ischemic tissue is re-perfused, causing tissue and cell damage. The initial ischemic stage disrupts blood supply to the organs, leading to metabolic imbalance and microvascular dysfunction. Upon the restoration of blood supply, sudden perfusion, and oxygenation exacerbate organ damage by activating innate and adaptive immune responses that produce free radicals [93]. IRI remains a significant challenge and a cause of graft failure [94], affecting organ function [95]. Transplants from marginal donors are susceptible to the detrimental effects of IRI and hypothermia [96]. Machine perfusion has emerged as a new strategy for preserving marginal grafts. Research by Cao et al. demonstrated that MSCs reduced inflammatory responses following donation-after-cardiac-death (DCD) liver transplantation compared with normothermic machine perfusion (NMP) alone, evidenced by lower levels of TNF-*α*, IL-1b, and IL-6, and the reduced expression of molecules associated with the HMGB1 and TLR4/NF-*k*B pathways [97]. Additionally, MSC-derived EVs can induce the expression of genes involved in anti-inflammatory response and oxidative stress resolution in rat lungs during NMP [98]. In animal liver transplantation models, MSCs have been demonstrated to reduce IRI [99].

Despite the availability of powerful immunosuppressants, acute allograft rejection is still common after liver transplantation. These drugs can decrease the body’s immunity, leading to complications such as malignant tumors and infections. Regarding malignant tumors, calcineurin inhibitors have been reported to activate the Rho/ROCK pathway, enhancing the migration of HCC cells [100]. Additionally, the migration and proliferation of cancer cells may be stimulated by calcineurin inhibitors via the VEGF, TGF-beta, and Ras pathways [100]. Immunotherapies hold significant promise in the treatment of various cancers. Specifically, the primary strategy for enhancing the antitumor response in HCC focuses on immune modulation. A recent study demonstrated that MSCs modified with in vitro-transcribed mRNA can be used to stimulate an antitumor response in HCC [101]. Regarding infections, studies have demonstrated that EVs secreted by MSCs reduce the pro-inflammatory factor IL-6 and the surface levels of CC chemokine ligand 7 while enhancing the expression of NOD-like receptor protein 12, which inhibits the inflammatory response. Umbilical cord MSC transfusion is feasible to treat acute graft rejection after transplantation and may mediate therapeutic immunosuppressive effects [102]. Additionally, studies have evaluated the efficacy of MSC therapy in patients with liver failure who received ABO-incompatible liver transplantation (ABO-iLT). The findings suggest that MSC therapy is as effective, if not more so, than rituximab in reducing the incidence of acute rejection. Patients treated with MSCs experienced lower rates of biliary complications and infections, suggesting that MSCs could serve as a novel immunosuppressive method for ABO-iLT [103].

There is a growing demand for noninvasive biomarker platforms to monitor immune rejection in liver transplantation. Evaluating graft quality, along with enhancing the phenotypic characterization of the pretransplant population, can build an early warning system for immunization. The preoperative assessment of factors such as primary disease and recipient age is crucial. Organ rejection can be categorized into three main types: ACR, antibody-mediated rejection, and chronic graft dysfunction. ACR results from adaptive immunity involving major histocompatibility complex (MHC) mismatch and T cell allogeneic recognition [98]. Besides alleviating IRI, MSCs aim to treat or prevent acute rejection [95]. Antibody-mediated rejection, triggered by the anti-donor human leukocyte antigen (HLA) or non-HLA antibodies, can occur independently or concurrent with ACR and is a major factor in chronic rejection, affecting recipient survival. EVs contain immune modulatory molecules, such as inhibitory molecules, cytokines, and growth factors. The packing of nucleic acids and other contents into EVs is regulated by signals from EVs themselves or the cellular/extracellular environment. Multiple studies have demonstrated that post-translational modifications can influence the selective loading of EV cargoes [104]. The packaging of RNA molecules into EVs is largely regulated by RNA binding proteins (RBPs), which recognize specific motifs within microRNAs. These RBPs can selectively prevent certain microRNAs from being incorporated into EVs by binding to particular sequences within the microRNAs [105]. For instance, under oxidative stress, the O-GlcNAcylation of the RBP hnRNPA2B1 increases its interactions with microRNAs secreted in EVs [106]. RNA packaging into EVs increases when the RBP hnRNPH is silenced [104]. Since biomarkers are easily detectable in biological fluids and reflect pathophysiologic conditions, variations in EV profiles can be used to predict and assess the severity of allograft rejection [107].

## 4. Organ Preservation and Restoration of Organ Function

To address donor shortages, one strategy is to utilize marginal grafts, including those from DCD donors or older donors [94,108]. To enhance the success of marginal donor transplantation, various studies are investigating innovative methods to improve graft quality. One promising approach is MSC-mediated therapy. Given the potential of MSCs and MSC-derived EVs in liver transplantation, numerous studies have explored their use alongside conventional drugs for marginal donor transplantation.

Immune cell infiltration is a critical factor leading to liver injury, accompanied by hepatocyte apoptosis, HSC activation, intrahepatic scar tissue formation, uncontrolled wound-healing processes, and tumorigenesis [109]. MSCs contribute to liver regeneration and repair by migrating to injured tissues, differentiating into hepatogenic cells, reducing hepatocyte apoptosis, promoting hepatocyte proliferation, and exerting immunoregulatory and anti-inflammatory effects. Additionally, the transplantation of MSCs and their derivatives promote liver regeneration and mitigate acetaminophen-induced liver injury [110,111]. Currently, EVs derived from MSCs and iPSCs through cell modifications enable targeted therapeutic delivery to organs before transplantation, thereby promoting tissue regeneration [38]. These modification techniques also offer opportunities to explore various therapeutic strategies, such as gene silencing and cell therapy, in fully functional grafts [112].

## 5. Current Research and Prospects

A recent controlled clinical study examined the impact of a single infusion of 1 × 10^6^/kg umbilical cord MSCs alongside conventional immunosuppressants for treating acute liver transplant rejection [113]. At a 12-week follow-up period, patients treated with MSCs demonstrated lower hepatic aminotransferase levels and histological improvements compared with the control group. There was an increase in circulating Treg cells and the Treg/Th17 ratio at 4 weeks. In cases of ischemic biliary lesions post-liver transplantation, 12 patients received six intravenous infusions of umbilical cord MSCs (1.0 × 10^6^/kg) on weeks 1, 2, 4, 8, 12, and 16. This treatment led to decreased alkaline phosphatase, total bilirubin, and *γ*-glutamyl transferase compared to the baseline. Moreover, 64.3% of the control group patients required interventional therapy, whereas only 33.3% of the MSC-treated group needed such interventions, reducing the necessity for additional therapy and improving the survival rates of the transplanted liver.

Optimizing the dosing of MSC therapy is crucial for its efficacy, and the infusion dose should be tailored to the severity of the patient’s condition. Current clinical trials typically utilize a single infusion dose of (1 to 2) × 10^6^/kg MSCs with a 4-week interval for reinjection, usually administering three to six injections as a treatment regimen. Key factors for high-quality stem cells include the source of cells, cell growth activity, and cell survival rate. The use of autologous bone marrow MSCs is generally not recommended because pathological conditions and donor age can greatly impact cell quality and therapeutic efficacy.

MSCs and MSC-derived EVs present innovative strategies for liver transplantation, including organ preservation, immune rejection therapy, targeted biological drug delivery, and biomarker detection. However, challenges remain in translating these therapies into clinical practice. Developing strategies to mitigate IRI is fundamental for successful transplantation, especially when using high-risk donors. Combining bioactive molecules like MSCs with machine perfusion has the potential to reduce organ damage. Nonetheless, the interactions between donor organs and recipients and the precise mechanism of MSC-mediated damage repair and regeneration are not yet fully understood. The continuous refinement of mechanical perfusion technology and its application standards is needed to enhance donor intervention and repair, especially when using marginal donors. Immune rejection is a primary cause of transplant failure, with treatment and prevention strategies currently including tissue matching, immune monitoring, and immunosuppression. While numerous studies have demonstrated the immunosuppressive effects of MSCs and MSC-derived EVs by modulating macrophages and T cells, the overall immune response involves a complex interplay between innate and adaptive immune responses. It is crucial to investigate whether MSC-derived EVs influence other immune cells, such as dendritic and natural killer cells, and how they regulate communication between various immune cells. In the context of liver transplantation, exosomes show promise as biomarkers due to their stability and ease of availability. However, their properties, contents, and distribution in complex in vivo and in vitro environments require further study. The clinical application of MSCs and MSC-derived EVs should focus on three main aspects: standardizing cell culture conditions, optimizing exosome isolation and purification methods, and identifying and monitoring their immunomodulatory effects in liver transplantation.

## 6. Conclusions

Improving organ availability and reducing immune rejection remain challenges in liver transplantation. Recent innovative strategies highlight the clinical potential of MSCs and EVs. Incorporating EVs into perfusion solutions can enhance the vitality and function of donor organs, mitigate IRI, and speed up postoperative tissue repair. MSCs can modulate rejection post-transplantation by influencing allogeneic recognition pathways and promoting donor-specific immune tolerance, presenting a promising approach for immunosuppression post-transplantation. Moreover, exosomes with specific recognition properties can serve as biomarkers and facilitate targeted drug delivery. MSCs, EVs, and exosomes hold substantial potential for future applications in liver transplantation.

## Figures and Tables

**Figure 1 cells-13-01604-f001:**
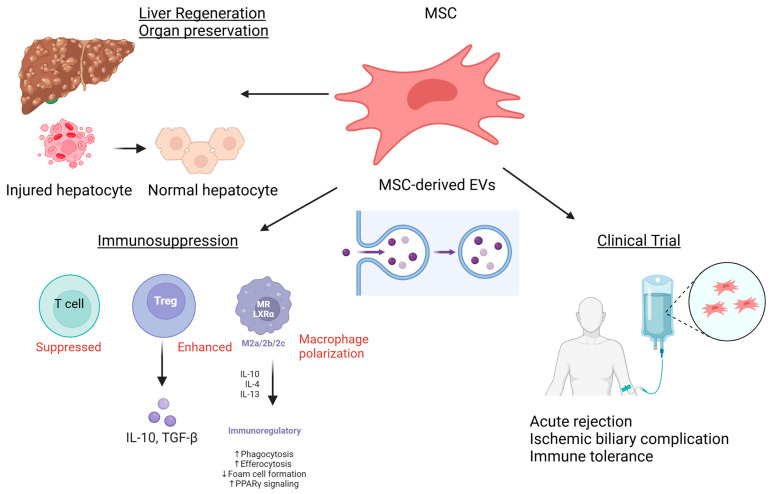
The role of MSCs in liver transplantation and liver regeneration.

## Data Availability

The original contributions presented in the study are included in the article, further inquiries can be directed to the corresponding author.

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
