# Peer review of "Innovative Strategies for Liver Transplantation: The Role of Mesenchymal Stem Cells and Their Cell-Free Derivatives"

_cells, 2024, doi:10.3390/cells13191604_

Round 1

Reviewer 1 Report

Comments and Suggestions for Authors

The condensed review on innovative strategies for liver transplantation is in my opinion a bit superficial, it lacks depth where this is needed in order to appreciate the novelty in comparison with other innovations in  this field. A few example are listed below

Title:

where is self-free in the title?

Introduction:

where are the adult or organ specific stem cells? Have may shift to may have (somewhere midway the introduction)What are MSCs?: first sentence please mention the tree lineage differentiation potential. How to define a hepatocyte-like cell, I know this term is often used but it is unclear what functions are expressed by that cell type.

MSCs in liver regeneration:

miRNAs, proteins, and metabolites

Third paragraph: unique proteins please be more specific and name a few. Same for varying transcriptome. First sentence of fourth paragraph lacks any reference. Referering to human umbilical cord-derived stem cells, does not seem to be an easy let alone autologous cell source.

Ischemia: complications such as malignant tumors and infections. Only the infection part is a further worked out. Last paragraph: signal from EVS, again please provide at least some examples

Author Response

- 1) The condensed review on innovative strategies for liver transplantation is in my opinion a bit superficial, it lacks depth where this is needed in order to appreciate the novelty in comparison with other innovations in this field. A few examples are listed below

Title:

where is self-free in the title?

Answer:

Thank you for your valuable feedback. The manuscript emphasizes both the utility of MSCs and cell-free therapies derived from MSCs (e.g., secretomes and EVs). Therefore, I agree that the title should be updated to reflect this. I have now added the term "Cell-Free" to the title.
<Modified part of the paper>

Title

Page 1 L1-2:

Title: Innovative Strategies for Liver Transplantation: The Role of Mesenchymal Stem Cells and Their Cell-free Derivatives

- 2) Introduction:

where are the adult or organ specific stem cells?

Answer:

Thank you for your suggestion. I have added a section on adult stem cells, including organ-specific stem cells, in the introduction.
<Modified part of the paper>

Introduction

Page 3 L14-18:

Among these, adult stem cells, including organ-specific stem cells such as hepatic progenitor cells, offer significant potential for liver regeneration.5 These cells can differentiate into specialized liver cells and have been shown to support liver repair. However, their regenerative capacity is more limited than embryonic stem cells (ESCs), induced pluripotent stem cells (iPSCs), and mesenchemal stem cells (MSCs).

- 3) Introduction:

Have may shift to may have (somewhere midway the introduction)

Answer:

I have made the suggested correction.
<Modified part of the paper>
Introduction
Page 3 L26-27:
Consequently, MSCs are extensively employed in tissue repair, regenerative medicine, and may improve transplant outcomes.

- 4)What are MSCs?:

first sentence please mention the tree lineage differentiation potential.

Answer:

I have now included the differentiation potential of MSCs into the three primary cell lineages.

 <Modified part of the paper>

Introduction

Page 4 L2-3:

MSCs are multipotent stem cells capable of differentiating into three primary cell lineages: adipogenic, chondrogenic, and osteogenic.

- 5)

How to define a hepatocyte-like cell, I know this term is often used but it is unclear what functions are expressed by that cell type.

Answer:

Thank you for your comment. I have now added more information about the functions of hepatocyte-like cells.
<Modified part of the paper>

Introduction

Page 4 L5-10: MSCs promote liver regeneration by differentiating into hepatocyte-like cells, which perform essential liver functions such as albumin synthesis, detoxification, and glucose metabolism.33,34 These hepatocyte-like cells are key to restoring liver function in damaged tissues. Furthermore, MSCs secrete bioactive factors that modulate immune responses, inhibit fibrosis, stimulate vascularization, and promote tissue remodeling.

-
6) MSCs in liver regeneration:

miRNAs, proteins, and metabolites

Third paragraph: unique proteins please be more specific and name a few. Same for varying transcriptome.

Answer:

Thank you for pointing this out. I have now included examples of unique proteins and specified different transcriptomes for MSCs derived from various sources.
<Modified part of the paper>

MSCs in Liver Regeneration

Page 6 L12-14: Unique proteins, such as heat shock protein 70 (HSP70) and tetraspanin CD63, have been identified in MSC-derived exosomes, distinguishing them from their parental MSCs.

Page 6 L15-19: For instance, adipose tissue-derived MSCs demonstrate more potent immunomodulatory effects compared to bone marrow-derived MSCs,57 while bone marrow-derived MSCs outperform adipose tissue-derived MSCs in preserving tissue viability and promoting neovascularization.

-
7) First sentence of fourth paragraph lacks any reference.

Answer:
Thank you for the observation. I have now included the appropriate reference.

<Modified part of the paper>

MSCs in Liver Regeneration

Page 6 L22-25: MSC-derived secretomes and exosomes have demonstrated significant potential in treating liver injuries by promoting hepatocyte proliferation, inhibiting apoptosis, necroinflammation, oxidative stress, immune rejection, and hepatic stellate cell (HSC) activation.59

-
8) Referring to human umbilical cord-derived stem cells, does not seem to be an easy let alone autologous cell source.

Answer:
You are absolutely right. I have added a section discussing the limitations of using human umbilical cord-derived stem cells.

<Modified part of the paper>

MSCs in Liver Regeneration

Page 7 L12-14: However, challenges remain, including the need for umbilical cord donation, manufacturing standardization, product characterization, and managing inter-donor variability.       

- 7) Ischemia: complications such as malignant tumors and infections. Only the infection part is further worked out.

Answer:

I appreciate your feedback. I have now expanded on the relationship between malignancy, immunosuppressants, and MSCs.
<Modified part of the paper>

Ischemia-Reperfusion Injury and Immunosuppression Therapy

Page 8 L22-30: Regarding malignant tumors, calcineurin inhibitors have been reported to activate the Rho/ROCK pathway, enhancing the migration of hepatocellular carcinoma cells.87 Additionally, the migration and proliferation of cancer cells may be stimulated by calcineurin inhibitors via the VEGF, TGF-beta, and Ras pathways.87 Immunotherapies hold significant promise in the treatment of various cancers. Specifically, the primary strategy for enhancing the antitumor response in hepatocellular carcinoma focuses on immune modulation. A recent study demonstrated that MSCs modified with in vitro transcribed mRNA can be used to stimulate an antitumor response in hepatocellular carcinoma.88

- 8) Ischemia: Last paragraph: signal from EVS, again please provide at least some examples

Answer:
I have added specific examples of signals from EVs.

<Modified part of the paper>

Ischemia-Reperfusion Injury and Immunosuppression Therapy

Page 9 L6-13: Multiple studies have demonstrated that post-translational modifications can influence the selective loading of EV cargoes. The packaging of RNA molecules into EVs is largely regulated by RNA binding proteins (RBPs), which recognize specific motifs within microRNAs. These RBPs can selectively prevent certain microRNAs from being incorporated into EVs by binding to particular sequences within the microRNAs.93 For instance, under oxidative stress, O-GlcNAcylation of the RBP hnRNPA2B1 increases its interactions with microRNAs secreted in EVs.92 RNA packaging into EVs increases when the RBP hnRNPH is silenced.

Reviewer 2 Report

Comments and Suggestions for Authors

The present review introduced the application of mesenchymal stem cells (MSCs) and extracellular vesicles (EVs) in liver regeneration and transplantation. The review also examines the mechanisms by which MSC-based cell-free strategies alleviate liver injury and how these innovations could improve outcomes in liver transplantation, addressing key challenges such as immune rejection and donor organ availability. There are several major points that are listed as below.

1. The major concern on the application of MSCs in liver regeneration is that whether the MSCs have the potential to contribute to the carcinogenesis. The authors should comprehensively discuss and address the point.

2. As the liver injury could be derived from different etiology including viral hepatitis, alcoholic liver disease, metabolic dysfunction associated steatotic liver disease and so on, are there difference in efficacy and mechanisms when utilizing the MSCs in these conditions?

3. The section “What are MSCs?” is basic knowledge that could be moved to introduction section.

Comments on the Quality of English Language

Moderate editing of English language required.

Author Response

Reviewer 2:

- 1. The major concern on the application of MSCs in liver regeneration is that whether the MSCs have the potential to contribute to the carcinogenesis. The authors should comprehensively discuss and address the point.

Answer:

Thank you for your comment. I have comprehensively addressed this concern in the liver regeneration section.
<Modified part of the paper>
MSCs in Liver Regeneration

Page 7 L28-42:

While MSCs have shown the ability to migrate and integrate into tumor tissues,74 their effects on HCC cells remain controversial. Zhao et al.75 found that adipose tissue-derived MSC-CM inhibited proliferation and promoted cell death in an HCC cell line in vitro. However, some research suggests that bone marrow-derived MSCs can enhance the migration and invasion of HCC cells.76,77 Studies have also raised concerns about the genetic instability and tumorigenic potential of MSC cultures.78 Rosland et al.79 reported that 45.8% of human MSCs underwent malignant transformation after extended culture periods. Ren et al.80 similarly observed that MSCs from adult cynomolgus monkeys transformed into highly tumorigenic mesenchymal cells after in vitro culture. Although long-term follow-up studies have indicated that MSC transplantation is generally safe, with no tumor formation reported in patients over 11 years and 5 months,81 it is still not clear how MSCs influence tumorigenesis and development in patients. further research is needed to clarify MSCs' potential role in tumor development. This highlights the need for continued evaluation and optimization of MSC-based clinical applications.

-2. As the liver injury could be derived from different etiology including viral hepatitis, alcoholic liver disease, metabolic dysfunction associated steatotic liver disease and so on, are there difference in efficacy and mechanisms when utilizing the MSCs in these conditions

Answer:

I appreciate your suggestion. I have now included examples of how MSCs work differently depending on the etiology.
<Modified part of the paper>
MSCs in Liver Regeneration

Page 6 L38-Page 7 L6:

Although there are various causes of liver injury, MSCs exhibit distinct mechanisms of action depending on the specific etiology.62 MSCs hold potential as a cellular therapy for patients with alcoholic liver disease by promoting differentiation, modulating the immune response, inhibiting liver fibrosis, and aiding in tissue regeneration. MSCs can differentiate into hepatic cells, helping to restore hepatocyte function lost due to alcohol consumption. They also induce the expression of hepatocyte markers, including cytokeratin 18 (CK18), CK19, and cytochrome P450 3A4, while enhancing glycogen storage and albumin secretion. Additionally, MSC therapy activates dendritic cells, lymphocytes, and Tregs.63 MSCs also release trophic factors such as VEGF, EGF, and IGF-1, which are involved in tissue repair processes.64 Regarding non-alcoholic steatohepatitis (NASH), recent studies have applied MSC-based therapy in mouse models that mimic the characteristics of NASH. Specifically, male immunodeficient mice were fed a high-fat diet for 21 weeks, followed by intrasplenic administration of 0.9-1 × 106 human bone marrow-derived MSCs.65,66 This treatment reduced liver fat deposition by threefold compared to control animals. Human mitochondria from the donor cells were detected in the mouse liver post-transplantation, contributing to lipid metabolism and resulting in a 25% reduction in triglyceride levels. Tissue inflammation and fibrosis were also alleviated, leading to restored tissue homeostasis.65,66

- 3. The section “What are MSCs?” is basic knowledge that could be moved to introduction section.

Answer:
Thank you for your valuable feedback. The purpose of this section is to provide an overview of MSCs for readers who may be unfamiliar with the concept. By presenting this information before delving into the main content, we aim to help readers understand what MSCs are, their functions, and what will be discussed in the following sections. I believe keeping this as a standalone section, along with the accompanying figure, enhances comprehension. Additionally, I have revised the section to further clarify MSC functions, which should deepen the reader's understanding before moving on to the core topics.

<Modified part of the paper>

What are MSCs?

Page 4 L1-21:

MSCs are multipotent stem cells capable of differentiating into three primary cell lineages: adipogenic, chondrogenic and osteogenic.9-15,30 MSCs are highly accessible for therapeutic use. MSCs offer several advantages in liver transplantation and regeneration (Figure 1).26,31,32 MSCs promote liver regeneration by differentiating into hepatocyte-like cells, which perform essential liver functions such as albumin synthesis, detoxification, and glucose metabolism.33,34 These hepatocyte-like cells are key to restoring liver function in damaged tissues. Furthermore, MSCs secrete bioactive factors that modulate immune responses, inhibit fibrosis, stimulate vascularization, and promote tissue remodeling.33 In the context of ischemia-reperfusion injury (IRI), MSCs can reduce inflammation and cell damage, improving graft survival. For immunosuppression therapy, MSCs help prevent acute cellular rejection (ACR) and graft-versus-host disease (GVHD) by modulating T-cell responses and increasing regulatory T cells (Tregs).32,35-37

Additionally, MSC-derived secretomes and EVs offer potential in organ preservation by enhancing the vitality and function of donor organs and mitigating IRI.38 MSC-derived secretomes and EVs also aid in the restoration of organ function by promoting tissue repair and reducing apoptosis and inflammation. These advantages position MSCs and their derivatives as promising therapeutic agents in liver transplantation, with the potential to improve outcomes and address challenges such as organ shortages and immune rejection.

Thank you for considering our revised manuscript.

Timothy M. Pawlik, MD, PhD, MPH, MTS, MBA, FACS, FSSO, FRACS (Hon.)

Professor and Chair, Department of Surgery

The Urban Meyer III and Shelley Meyer Chair for Cancer Research

Professor of Surgery, Oncology, and Health Services Management and Policy

Surgeon in Chief, The Ohio State University Wexner Medical Center

The Ohio State University, Wexner Medical Center

Round 2

Reviewer 2 Report

Comments and Suggestions for Authors

The authors have already addressed the points raised by the reviewers in the revised manuscript. The manuscript could be considered for publication.

Comments on the Quality of English Language

Minor editing of English language required.